# Evolution of the head-trunk interface in tetrapod vertebrates

Elizabeth M Sefton[1,2]*, Bhart-Anjan S Bhullar[1,2,3,4,5], Zahra Mohaddes[1,2], James Hanken[1,2]

[1]Department of Organismic and Evolutionary Biology, Harvard University, Cambridge, United States; [2]Museum of Comparative Zoology, Harvard University, Cambridge, United States; [3]Department of Organismal Biology and Anatomy, University of Chicago, Chicago, United States; [4]Department of Geology and Geophysics, Yale University, New Haven, United States; [5]Yale Peabody Museum of Natural History, Yale University, New Haven, United States

**Abstract** Vertebrate neck musculature spans the transition zone between head and trunk. The extent to which the cucullaris muscle is a cranial muscle allied with the gill levators of anamniotes or is instead a trunk muscle is an ongoing debate. Novel computed tomography datasets reveal broad conservation of the cucullaris in gnathostomes, including coelacanth and caecilian, two sarcopterygians previously thought to lack it. In chicken, lateral plate mesoderm (LPM) adjacent to occipital somites is a recently identified embryonic source of cervical musculature. We fate-map this mesoderm in the axolotl (*Ambystoma mexicanum*), which retains external gills, and demonstrate its contribution to posterior gill-levator muscles and the cucullaris. Accordingly, LPM adjacent to the occipital somites should be regarded as posterior cranial mesoderm. The axial position of the head-trunk border in axolotl is congruent between LPM and somitic mesoderm, unlike in chicken and possibly other amniotes.

**\*For correspondence:** esefton@oeb.harvard.edu

**Competing interests:** The authors declare that no competing interests exist.

## Introduction

The evolution of a mobile neck was a key innovation at the origin of tetrapods (*Daeschler et al., 2006*). It involved expansion of muscles, some derived from the head (cranial muscles) and some from the trunk, to support the skull apart from the pectoral girdle and permit a greater range of movement of the head relative to the rest of the body. Cranial muscles support a variety of functions, including feeding, respiration, vision, facial expression and vocalization. They are distinct from trunk muscles in genetic regulation and susceptibility to disease (*Noden, 1983*; *Noden et al., 1999*; *Sambasivan et al., 2009*; reviewed by *Bismuth and Relaix, 2010*; *Diogo et al., 2015*; *Noden and Francis-West, 2006*; *Tzahor, 2009*). Developmentally, they are non-somitic, arising instead from cranial paraxial and splanchnic mesoderm (*Couly et al., 1992*; *Noden, 1983*; *Evans and Noden, 2006*; reviewed by *Noden and Trainor, 2005*). Cranial muscle regulatory factors include *Isl1, Tbx1, MyoR, Capsulin* and *Pitx2*, which operate in specific muscle groups (*Hacker and Guthrie, 1998*; *Sambasivan et al., 2009*; *Lu et al., 2002*; *Mootoosamy and Dietrich, 2002*; *Harel et al., 2009*). *Pitx2*, for example, specifies mandibular arch mesoderm but not hyoid arch mesoderm in the mouse (*Shih et al., 2007a*). In contrast to cranial muscle, formation of trunk muscle is *Pax3*-dependent (*Tajbakhsh et al., 1997*).

The domain of the vertebrate neck contains two muscle groups: the hypobranchial muscles ventrally and the cucullaris dorsally. Hypobranchial muscles are derived from occipital somites, which form the hypoglossal cord and migrate towards the tongue (*Noden, 1983*; *O'Rahilly and Müller, 1984*). The number of occipital somites contributing to cranial structures varies among species,

**eLife digest** Muscles in the head and trunk (main body) form from different parts of the embryo, and their development uses different genes. Trunk muscles are derived from somites – paired blocks of cells arranged in segments on either side of the midline (which divides the body into left and right halves). By contrast, cells that give rise to head muscles are arranged in a continuous mass.

But what about neck muscles? Some studies claim they develop like head muscles; others suggest they are trunk muscles. These studies commonly examine mice or chickens. By examining species that have a more primitive complement of head and neck muscles, Sefton et al. now show that a neck muscle should be considered a kind of head muscle.

Gill muscles are definitive head muscles. Sefton et al. found that the cucullaris, a prominent neck muscle in fishes and amphibians, forms from the same mass of cells that gives rise to gill muscles. Moreover, studying muscle development in Mexican axolotls showed that cells that contribute to gill muscles extend into the trunk, which is further back in the embryo than was previously known.

Previous studies reported the cucullaris muscle is absent in a "lobe-finned" fish called the coelacanth, which is closely related to four-limbed animals. However, by using a technique called micro-computed tomography to visualize the neck muscles of this fish, Sefton et al. show that the cucullaris muscle is present and connects the rear-most gill to the shoulder.

The finding that neck muscles form like head muscles in the axolotl confirms a previous claim that was based on studies of bird embryos. A future challenge is to understand the molecular and genetic mechanisms that establish the boundary between head and trunk muscles, and work out how those mechanisms might have influenced how the neck evolved.

however. For example, somites 2 and 3 form both hypobranchial musculature and the occipital arch in the axolotl (*Piekarski and Olsson, 2007*; *2014*), whereas in chicken somites 2–5 form both the occipital region of the skull and tongue musculature (*Couly et al., 1993*; *Huang et al., 1999*; *2000*).

The cucullaris muscle, a feature of gnathostomes, connects the head to the pectoral girdle, thus spanning the transition zone between cranial and trunk myogenic signaling regimes (*Kuratani, 1997*). It is the putative homologue of the trapezius and sternocleidomastoid in amniotes (*Lubosch et al., 1938*). In sharks and the Queensland lungfish, the cucullaris elevates the gill arches and protracts the pectoral girdle. It originates near the skull and continues caudally and ventrally to insert on the scapular region of the pectoral girdle; a ventral fascicle extends to the posteriormost branchial bar (*Edgeworth, 1926*; *1935*; *Allis, 1917*; *Vetter, 1874*; *Greenwood and Lauder, 1981*). The cucullaris is a thin muscle, and it can be difficult to visualize its three-dimensional position vis-à-vis adjacent skeleton and musculature. Hence, it is poorly described in many taxa with regard to both its shape and its relation to other cranial and trunk musculature. It is innervated by the accessory ramus of the vagus (X) nerve in anamniotes, but primarily by the accessory (XI) nerve in amniotes (*Edgeworth, 1935*). While in chicken the connective-tissue component of hypobranchial muscles and the ventrolateral neck region is derived from neural crest (*Le Lièvre and Le Douarin, 1975*), the cucullaris is reported to have somite-derived connective tissue (*Noden, 1983*). The derivation of connective-tissue components of the mouse trapezius is not fully resolved; both lateral plate mesoderm (*Durland et al., 2008*) and neural crest (*Matsuoka et al., 2005*) are reported sources.

While a somitic derivation of the hypobranchial muscles is widely accepted, the embryonic origin of the cucullaris is controversial (reviewed by *Tada and Kuratani, 2015*; *Ericsson et al., 2013*). Historically, the cucullaris was considered a branchiomeric cranial muscle based in part on its anatomical relation to the gill levators (*Vetter, 1874*; *Edgeworth, 1935*; *Piatt, 1938*). Subsequent fate mapping of anterior somites in chicken and axolotl, though, demonstrated a somitic (trunk) contribution (*Noden, 1983*; *Couly et al., 1993*; *Huang et al., 1997*; *2000*; *Piekarski and Olsson, 2007*). More recent fate mapping in chicken and genetic analysis in mouse reveal that the trapezius is primarily a lateral plate mesoderm-derived structure that employs a cranial, rather than trunk, myogenic program (*Theis et al., 2010*; *Lescroart et al., 2015*).

These data leave unresolved whether the lateral plate origin of the cucullaris is the result of a posterior shift of the head myogenic program or if instead head mesoderm extends caudally into the region adjacent to the anterior somites. To distinguish between these hypotheses, it is important to define the posterior limit of myogenic cranial mesoderm in an organism with a relatively conservative cervical and branchial region. Amniote branchial-arch musculature is reduced in comparison to that of piscine sarcopterygians and aquatic salamanders such as the axolotl, which has a relatively plesiomorphic arrangement of cranial muscle. Moreover, axolotls possess bushy external gills and their associated musculature, which likely were present in the larvae of Paleozoic tetrapods, as well as a robust gill skeleton, which was present in the earliest limbed stem tetrapods (*Schoch and Witzmann, 2011*).

Here, we address this problem from a combined morphological, genetic and developmental perspective. In the axolotl, we locate the head-trunk boundary within unsegmented cranial mesoderm. In addition, we use micro-computed tomography (CT) to describe the morphology of the cucullaris and gill levators in a phylogenetically diverse series of gnathostome taxa, including limbless caecilians and the coelacanth, the sister taxon to all other extant lobe-finned fishes. In previous studies, the cucullaris was investigated largely by gross dissection. In many species, however, the cucullaris is a thin, superficial muscle embedded in several layers of fat and connective tissue. It can be difficult to expose without damaging its in situ context with respect to the trunk and the pectoral girdle. Our CT-based reconstructions reveal such three-dimensional relationships without tissue disruption.

In those taxa that have both branchial levators and a cucullaris, the cucullaris consistently appears to be in series with the levators. This suggests that the cucullaris is a serial homolog of the levators, thus supporting a cranial muscle identity of the cucullaris. Likewise, in the axolotl, although the cucullaris in adults assumes a large, triangular 'trapezius-like' morphology, the larval cucullaris is clearly in series with the levators. The ubiquity of the cucullaris further supports the hypothesis that it is a critical component of the head-trunk connection in gnathostomes. To study the development of tissues in the transitional region spanned by the levators and cucullaris, we extend modern fate-mapping techniques and gene-expression analysis of cranial mesoderm to the axolotl. We show that unsegmented mesoderm adjacent to the anterior three somites contributes to the cucullaris as well as to the gill-levator muscles in a manner consistent with their apparent serial homology, which supports categorization of the cucullaris as a branchiomeric muscle. Cranial mesoderm markers, including *isl1* and *tbx1,* also are expressed in the developing cucullaris region. We find molecular regionalization of the cranial muscles at stage 40, with distinct expression patterns in the mandibular and hyoid arch musculature. Adductor muscles within the mandibular arch have distinct gene expression patterns as well. We argue that the posterior limit of cranial mesoderm in the axolotl extends caudally to the axial level of somite 3 and that the head-trunk boundary is consistent between the somites and lateral plate mesoderm. We discuss the importance of posterior cranial mesoderm in the evolution of the vertebrate neck.

## Results

### Morphology and conservation of cranial muscle

New soft-tissue-contrast staining methods for high-resolution CT afforded us the opportunity to examine the volumetric anatomy of muscles in a sample of vertebrates spanning Gnathostomata (*Figure 1*; *Video 1*; *Figure 1—figure supplements 1–8*). In Chondrichthyes, such as the chimaera, the cucullaris is a massive muscle that may incorporate anterior gill levators; in this respect, it may not be strictly homologous, in its entirety, with the cucullaris of osteichthyans. In piscine osteichthyans, such as bichir and lungfish, the cucullaris is a thin, strap-like muscle, sometimes called the 'protractor pectoralis' (e.g., *Greenwood and Lauder, 1981*). It diverges from the anterior gill levators, but otherwise it is in series with them, is located in the same connective tissue sheath, and shares muscle fibers with the immediately anterior levators. It originates from the posterior region of the head and inserts on the pectoral girdle and, when present, the fifth gill arch. In some amphibians and in amniotes, the cucullaris is a large wedge-shaped muscle, sometimes termed the trapezius (*Owen, 1866*; *Edgeworth, 1935*; *Gegenbaur et al., 1878*). This morphology is seen clearly in an *Anolis* lizard and in an opossum, which exhibits the primitive mammalian condition (*Figure 1F*; *Figure 1—figure supplements 7,8*).

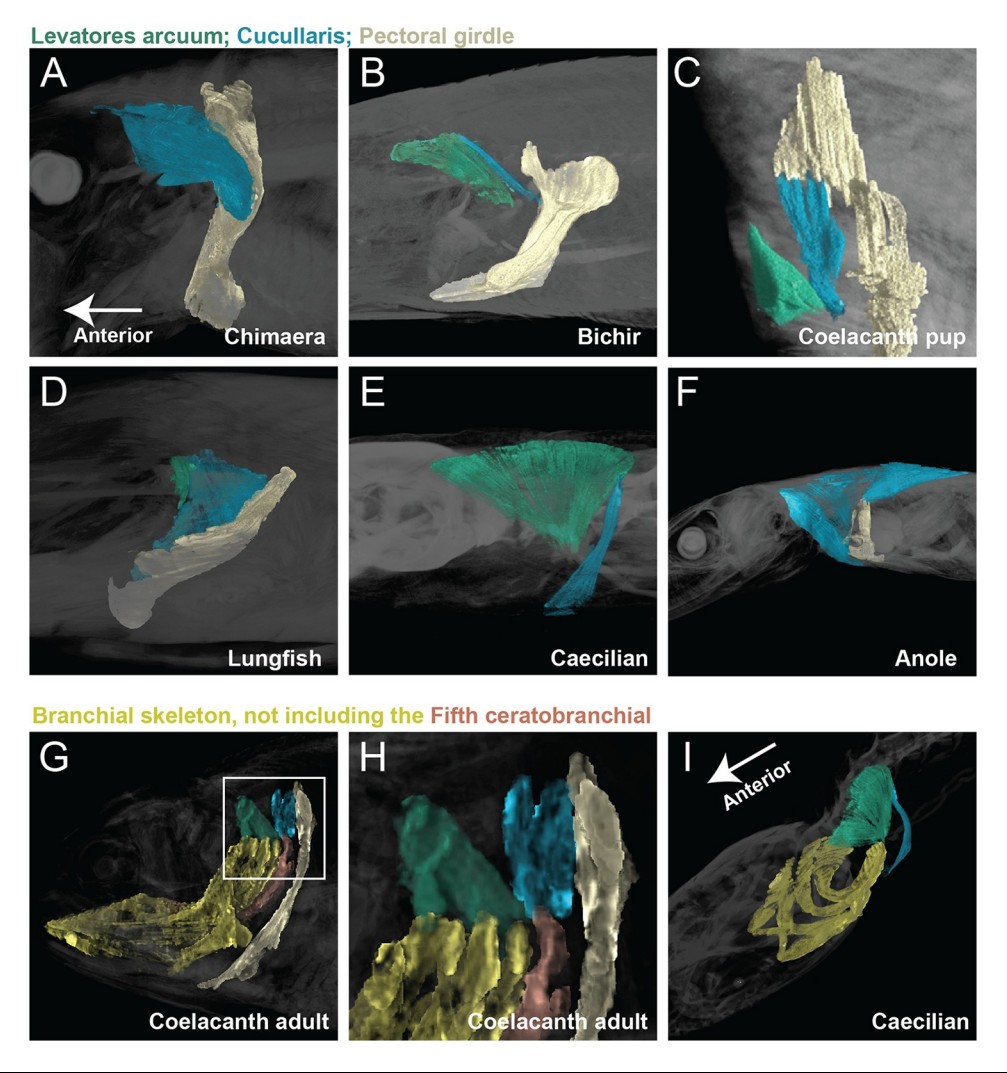

**Figure 1.** Cranial muscle evolution based on contrast-stained CT scans and an MRI scan (coelacanth adult). (A–F) Left lateral views of gill-levator musculature and the cucullaris (or its homologue) in representative gnathostomes, showing its insertion on the pectoral girdle (except in caecilians, where it inserts on ventral fascia). (G, H) Left lateral views of gill levators and the cucullaris in relation to the branchial skeleton in a coelacanth. The cucullaris attaches to the posteriormost gill arch. Box in **G** is enlarged in **H**. (I) Left dorsolateral view of the cucullaris in a caecilian. The gill-levator musculature is shaded green, the cucullaris blue, and the pectoral girdle white. In the lower panels, the fifth ceratobranchial is in pink and the anterior branchial skeleton in yellow.

The following figure supplements are available for figure 1:

**Figure supplement 1.** Stereo image of chimaera from *Figure 1* with skeletal elements and muscles segmented.

**Figure supplement 2.** Stereo image of bichir in dorsolateral view.

**Figure supplement 3.** Stereo image of lungfish in dorsolateral view.

**Figure supplement 4.** Stereo image of coelacanth in dorsolateral view.

**Figure supplement 5.** Stereo image of axolotl in dorsolateral view.

**Figure supplement 6.** Stereo image of caecilian in dorsolateral view.

*Figure 1 continued*

**Figure supplement 7.** Stereo image of anole in dorsolateral view.

**Figure supplement 8.** Stereo image of opossum in lateral view.

The cucullaris has not been described in the musculoskeletally conservative coelacanth, nor in the limbless caecilian amphibians. Based on novel dissections, *Greenwood and Lauder (1981)* reported the cucullaris absent from the coelacanth. *Millot and Anthony (1958)*, however, had earlier briefly described in the coelacanth a fifth gill levator that originates on the anocleithrum of the pectoral girdle, unlike the first four gill levators, which originate in the otic region. We examined this muscle in both a CT scan of a contrast-stained coelacanth pup and an MRI (magnetic resonance imaging) scan of an adult. The muscle in these specimens is larger than previously described, with several heads originating on the pectoral girdle (*Figure 1C*). It is angled differently from the other levators, but its fibers remain in close association with them and extend from the anocleithrum to insert on the fifth ceratobranchial. In the adult, an anterior portion of this muscle extends dorsally to attach on the fascia of the epaxial musculature (*Figure 1G,H*). Based on its morphology and location, we regard the fifth gill levator as the homolog of the cucullaris. Accordingly, the coelacanth cucullaris retains the ancestral connection between the posteriormost branchial bar and the pectoral girdle, which is seen in at least some sharks and lungfish (*Edgeworth, 1926*; *1935*; *Allis, 1917*; *Vetter, 1874*; *Greenwood and Lauder, 1981*). Even though actinopterygians lack an ossified fifth gill arch, the cucullaris in these taxa sometimes joins the fibers of the posteriormost gill levator (*Greenwood and Lauder, 1981*). Although the cucullaris in the coelacanth does not attach to the head, it retains a division that extends rostrally, terminating upon the dorsal fascia posterior to the cranium; it also retains the ancestral connection between the pectoral girdle and the cranial skeleton by its attachment to the branchial skeleton.

In the caecilian *Typhlonectes natans* we examined the m. levator arcus branchiales complex, previously described in *Dermophis mexicanus* (*Bemis et al., 1983*). The muscle is also termed the m. cephalodorsosubpharyngeus (*Wilkinson and Nussbaum, 1997*; *Lawson, 1965*). Based on our examination, the m. levator arcus branchiales complex is a triangular structure that originates from the otic capsule and dorsal trunk muscle fascia and inserts ventrally on the the posteriormost ceratobranchial (*Figure 1E,I*). A posterior division of the muscle, the pars posterosuperficialis (*Wilkinson and Nussbaum, 1997*), inserts on the fascia separating the rectus abdominus from the interhyoideus.

Based on these topographic relationships, we homologize the posterior division of the m. levator arcus branchialis complex with the cucullaris. It is unclear if a portion of the anterior division should also be considered part of the cucullaris, connecting to the adult hyobranchial skeleton.

The cucullaris and its homologs comprise a highly conserved connection between head and trunk. In general, the cucullaris is intimately associated, and sometimes partially continuous, with the gill levators. In numerous taxa, it attaches to the skull and gill skeleton, both cranial elements.

## Mesoderm adjacent to anterior somites forms the cucullaris and gill levators

In amphibians, the cucullaris has also been termed the protractor pectoralis (*Ziermann and Diogo, 2013*) and the trapezius (*Piatt, 1938*). In juvenile axolotls, the cucullaris resembles the condition in bichir; it is morphologically similar to and in series with the anterior gill levators,

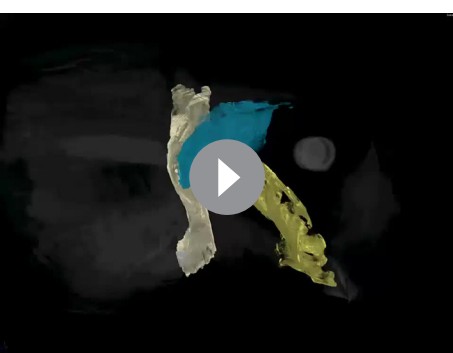

**Video 1.** Video of representative gnathostome cranial and pectoral regions spinning around their long axes with skeletal elements and muscles segmented. See *Figure 1* for color guide. Videos rendered in VGStudio Max v2.2 at 2048 X 1536 resolution, 25 frames per second (20 s), Windows AVI format, 85% quality. Sides segmented represent original specimens, whereas some images in figure panels were reversed so that all specimens are oriented in the same direction.

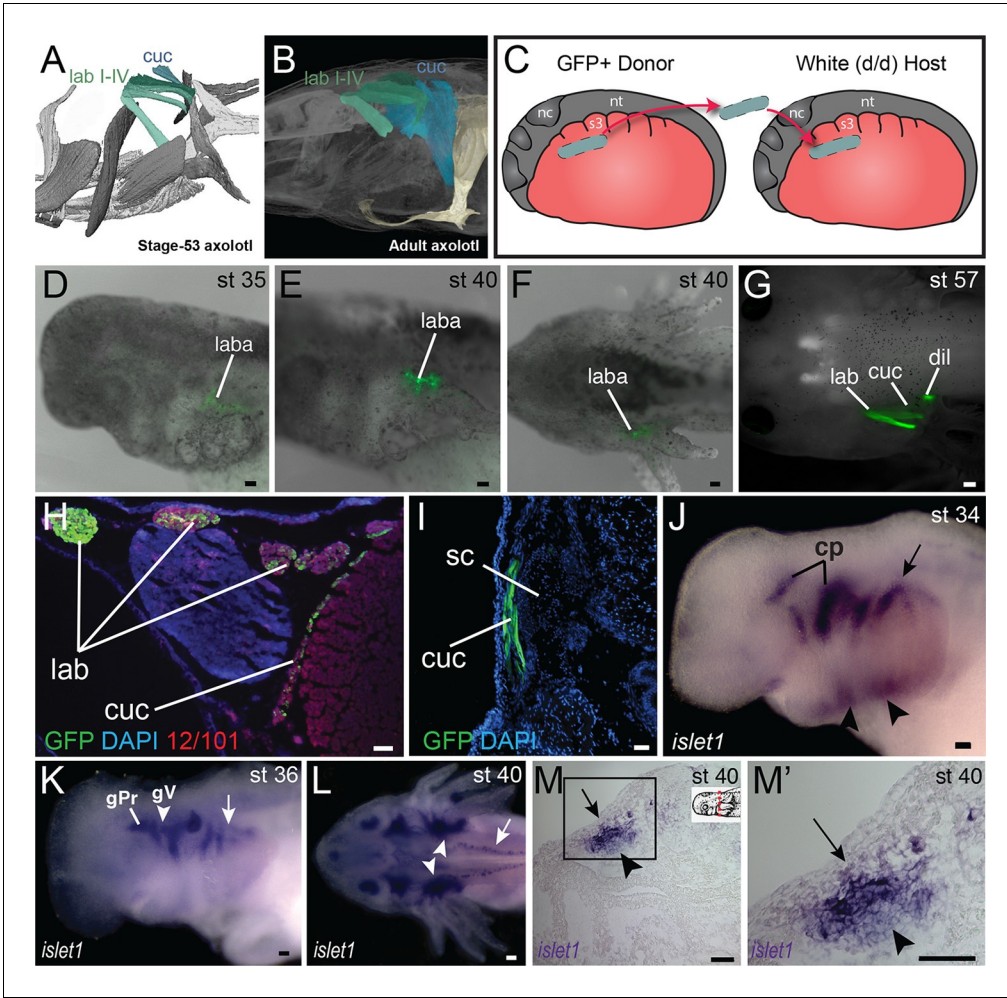

**Figure 2.** Development of the cucullaris muscle in the axolotl. (**A**, **B**) Morphology of the developing cucullaris, with the four gill-levator muscles (lab I–IV) shaded light green and the cucullaris (cuc) blue. More-posterior muscles are shaded dark green. Anterior is to the left. (**A**) Dorsolateral view of an OPT scan of a juvenile axolotl stained with the 12/101 muscle antibody. (**B**) Contrast-stained CT scan of an adult axolotl in lateral view. The cucullaris is expanded into a broad sheet that inserts on the scapula. (**C**) Schematic depiction of an orthotopic transplantation of unsegmented mesoderm lateral to somites 1−3 at stage 21. Lateral views; anterior is to the left. nc, neural crest; nt, neural tube; s3, somite 3. (**D**−**I**) GFP labeling following stage-21 transplantation of unsegmented mesoderm lateral to somites 1−3. (**D**−**F**) Labeling of the levator arcuum branchiarum anlagen (laba) dorsal to the developing gills is visible in lateral (**D**, **E**) and dorsal (**F**) views. Anterior is to the left. (**G**) Gill-levator muscles (levator arcuum branchiarum, lab) of arches 3 and 4, the cucullaris (cuc) and the dilatator laryngis (dil) are labeled in a juvenile axolotl. Dorsal view; anterior is to the left. (**H**, **I**) Transverse sections through the posterior occipital region (**H**) and anterior trunk (**I**) of a juvenile axolotl. GFP labeling is visible in the gill levators and anterior cucullaris (**H**) and in the posterior cucullaris near its attachment with the scapula (**I**; sc). Lateral is to the left; dorsal is to the top. (**J**−**M'**) *isl1* expression in albino embryos. (**J**) At stage 34, *isl1* is expressed in ventral mesoderm, in the developing heart region (arrowheads) and around the dorsal cranial placodes (cp). Arrow indicates several stripes of expression dorsal to the developing gills. (**K**) At stage 36, *isl1* marks the profundal (gPr)/trigeminal (gV) placode region and earlier expression is maintained dorsal to the gills (arrow). (**L**) At stage 40, *isl1* is expressed in neurons within the dorsal spinal cord (arrow) and in the gill-levator region (arrowheads). Dorsal view. (**M**, **M'**) Transverse section of a stage-40 embryo with *isl1* expression in the dorsal gill levator region (arrows) and ganglia (arrowheads). Box in M is enlarged in M'. Scale bars, 100 µm, except G, 500 µm.

The following figure supplements are available for figure 2:

**Figure supplement 1.** Additional stages of embryonic *isl1* expression in *A. mexicanum*.

*Figure 2 continued*

**Figure supplement 2.** Embryonic expression of *tbx1* and *msc* in *A. mexicanum*.
**Figure supplement 3.** Mesoderm fate mapping in *A. mexicanum* embryos.

whereas in adults it expands into a broad, thin sheet (*Figure 2A,B*). Given the conservative morphology of branchiomeric musculature in the axolotl (*Ericsson et al., 2004*; *Ericsson and Olsson, 2004*; *Ziermann and Diogo, 2013*), we began fate-mapping head mesoderm that contributes to the pharyngeal arches. In chicken and axolotl, mesoderm from somites 1 and 2 contributes to the cucullaris (*Piekarski and Olsson, 2007*). In chicken, however, the majority of the cucullaris is derived from lateral plate mesoderm adjacent to the occipital somites (*Theis et al., 2010*). Consequently, we suspected that the axolotl cucullaris might also have a dual origin from both somitic and unsegmented mesoderm.

We transplanted GFP+ mesoderm adjacent to the first three somites into a white (d/d) host (*Figure 2C*). By stage 35, GFP+ cells were visible dorsal to the developing gill buds in the region of the presumptive gill muscles, the cucullaris and the dilatator laryngis (*Figure 2D−G*). In cross section, labeled cells were present throughout the length of the cucullaris but absent from the somitic hypobranchial or epaxial muscles, thus indicating little if any somitic contamination (*Figure 2H,I*). Additional transplantations were then performed for three regions of cranial mesoderm anterior to the first somite. In most of these transplants, the first gill-levator muscle originated from mesoderm just anterior to the first somite, whereas the posterior three gill levators arose from unsegmented mesoderm at the level of somites 1−3 (*Figure 2—figure supplement 3*).

## *Isl1* is expressed in the gill levator and cucullaris region

Transcription factors involved in cranial muscle development are expressed in gill levator/cucullaris muscle territory. At neurula and tailbud stages, *isl1* is expressed in anterior cranial mesoderm associated with the second heart field (*Figure 2—figure supplement 1*; *Sefton et al., 2015*). In subsequent stages, *isl1* expression expands dorsally to encompass the entire dorsoventral length of the gills (*Figure 2—figure supplement 1D*), but later (stage 34) it is reduced near the heart and appears in the developing cranial placodes (*Figure 2J*). From stage 36 through at least stage 40, stripes of expression are present dorsal to the developing gills, including the levator anlage (*Figure 2K−M'*). The cranial-mesoderm marker *tbx1* (sequences of *tbx1*, *msc* and *pitx2* by pers. comm. from J. Whited, B. Haas and L. Peshkin) is expressed in the developing gill muscle region at stages 35 and 38 (*Figure 2—figure supplement 2I*). By stage 38, *msc* is also expressed in the gill region. Unlike *isl1*, expression of *tbx1* and *msc* extends distally into the external gills (*Figure 2—figure supplement 2O*).

## Molecular regionalization of mandibular and hyoid arch muscles

In axolotl, jaw adductor muscles include the levator mandibulae externus (lme) and the levator mandibulae anterior (lma); the latter muscle is also called the pseudotemporalis (*Ziermann and Diogo, 2013*). Both of these muscles develop within the mandibular arch (*Figure 3B,I*). We examined expression of *lhx2*, a LIM-domain transcription factor involved in pharyngeal muscle specification in the mouse (*Harel et al., 2012*). As seen in mouse and *Xenopus*, *lhx2* is expressed in axolotl in the brain and eye (*Figure 3*; *Figure 4—figure supplement 1A−E*; *Viczian et al., 2006*; *Atkinson-Leadbeater et al., 2009*). At stage 34, *lhx2* is expressed in the mesodermal core of the pharyngeal arches (*Figure 3E,F*), but by stage 40 it becomes more restricted to specific muscle groups, including the lme and ventral hyoid arch musculature, but expression was not visible in the lma at stage 40 (*Figure 3G,H*). At stage 38, the anlage of the lma is located posterior to the eye (*Figure 3I*). By stage 40, this region expresses *isl1*, including the superficial lma and developing ganglia, while *isl1* expression was not visible in the lme (*Figure 3J−K''*).

A third gene provides an additional example of genetic heterogeneity in cranial muscle development. In mouse, *Pitx2* is broadly expressed in developing muscle; it is required to specify mesoderm of the mandibular arch but not of the hyoid arch (*Shih et al., 2007a*; *2007b*). In axolotl, as in chick, *pitx2* is expressed in anterior ectoderm and oral region from neurula stages

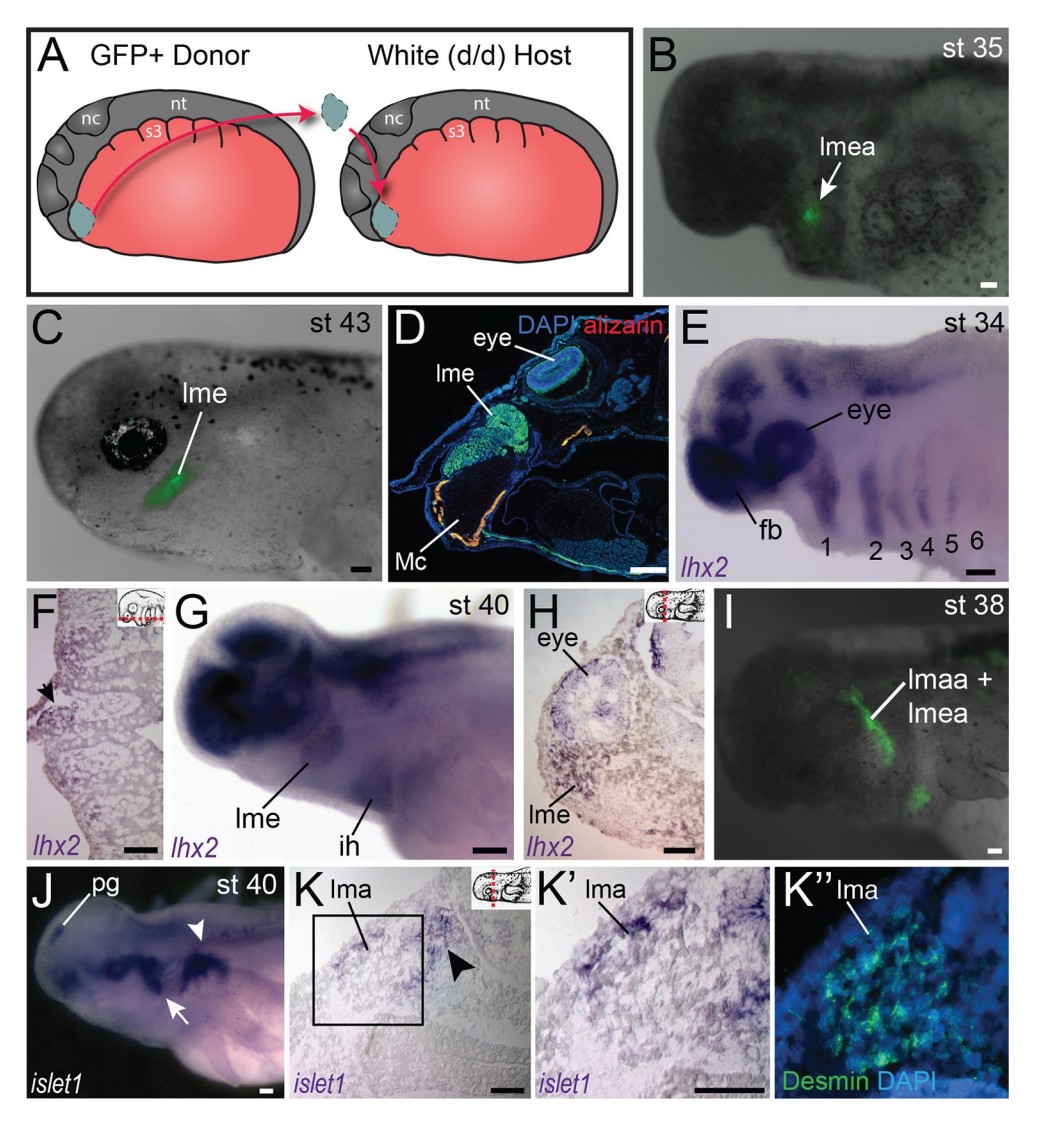

**Figure 3.** Fate-mapping and gene expression in mandibular adductor muscles. (**A**) Schematic depiction of orthotopic transplantations of anterior cranial mesoderm. nc, neural crest; nt, neural tube; s3, somite 3. (**B**) Labeling of mandibular arch mesoderm at stage 35 following transplantation at stage 19. Arrow points to the anlage of the levator mandibulae externus (lmea). (**C**) Specimen in (**B**) at stage 43, with labeling of the levator mandibulae externus (lme). (**D**) Transverse section through the eye region of a stage-55 axolotl following transplantation at stage 20. The levator mandibulae externus is labeled ventral to the eye. Mc, Meckel's cartilage. Lateral is to the left; dorsal is to the top. (**E**) At stage 34, *lhx2* is expressed in the pharyngeal mesoderm of all six arches (1–6) as well as the forebrain (fb) and eye. (**F**) Frontal section through the head at stage 36 showing *lhx2* expression in the mesodermal core of the third pharyngeal arch (arrow). Anterior is to the top. Inset panel depicts plane of section (dashed red line). (**G**) At stage 40, *lhx2* is expressed in the levator mandibulae externus (lme) and in the interhyoideus (ih), a ventral cranial muscle. (**H**) Transverse section at the level of the eye at stage 40, showing *lhx2* expression in the levator mandibulae externus. Dorsal is to the top. (**I**) Labeling of mandibular and hyoid arch mesoderm in a stage-38 embryo, including the anlage of the levator mandibulae anterior (lmaa), an anterior jaw adductor. Lateral view; anterior is to the left. (**J–K"**) *isl1* expression in albino embryos. (**J**) At stage 40, *isl1* is expressed dorsal to the gills (arrowhead) and in the pineal gland (pg). Expression posterior to the eye (arrow) overlaps with the region forming the levator mandibulae anterior. (**K–K"**) Transverse sections of a stage-40 embryo. Box in **K** is enlarged in **K'** and **K"**. (**K**) *isl1* is expressed in the lateral portion of the developing levator mandibulae anterior (lma) and in the trigeminal nerve (arrowhead). (**K"**) Desmin staining of muscle cells, including those expressing *isl1*. Scale bars, 100 µm.

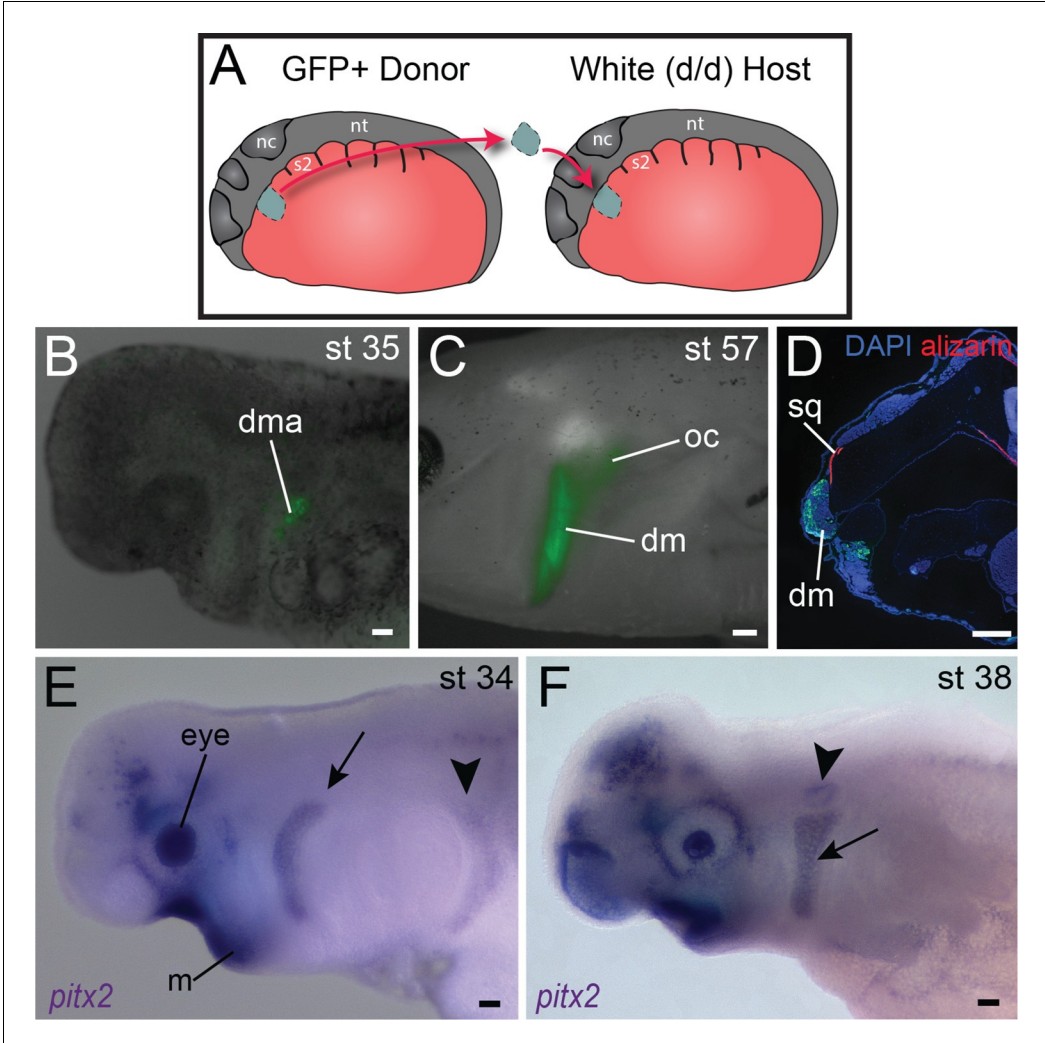

**Figure 4.** Origin of the mandibular depressor muscle and expression of *pitx2* in the hyoid arch. (A) Schematic depiction of orthotopic transplantation of cranial mesoderm. nc, neural crest; nt, neural tube; s2, somite 2. Somite 1 is small and triangular in shape. (B) GFP labeling of dorsal hyoid-arch mesoderm at stage 35 following transplantation at stage 20 includes the anlage of the depressor mandibulae (dma). (C) Specimen in (B) at stage 57, with labeling of the depressor mandibulae (dm) and otic capsule (oc). (D) Labeling of the depressor mandibulae in a transverse section through the jaw region of a stage-57 juvenile axolotl. sq, squamosal bone. Dorsal is to the top; lateral is to the left. (E) At stage 34, *pitx2* is expressed in the eye, the ventral mandibular arch (m), the hyoid arch (arrow) and more faintly in migrating somitic cells (arrowhead). (F) At stage 38, *pitx2* expression is maintained in the hyoid arch (arrow) and is also present in the otic vesicle (arrowhead). Scale bars, 100 µm, except C, 500 µm.

The following figure supplement is available for figure 4:

**Figure supplement 1.** Embryonic expression of *lhx2* and *pitx2* in *A. mexicanum*.

through at least tailbud stages (*Figure 4E,F*; *Figure 4—figure supplement 1F–M*; *Bothe and Dietrich, 2006*). A stripe of expression in hyoid arch mesoderm and in the migrating hypobranchial muscle precursors also appears by stage 34 (*Figure 4E*). It is maintained in hyoid arch derivatives and by stage 40 is concentrated in the hyoid musculature, including the depressor mandibulae and branchiohyoideus externus (*Figure 4F*; *Figure 4—figure supplement 1K,M*). The depressor mandibulae anlage (dma) is in the dorsal/proximal pharyngeal arch at stage 35 (*Figure 4B,C*) and then extends to insert on Meckel's cartilage at stage 42 (*Ericsson and Olsson,*

*2004*). At stage 40, *pitx2* is not expressed in gill musculature, although it is strongly expressed in tongue musculature (*Figure 4—figure supplement 1L,M*).

## Heterotopic transplantation between lateral plate and cranial mesoderm

We investigated the myogenic properties of lateral plate mesoderm adjacent to the somites to determine if local signals in the cranial mesoderm of the mandibular and hyoid arch regions could instruct lateral plate mesoderm at various axial levels to adopt cranial muscle fate. Myogenic lateral plate mesoderm in the cucullaris region adjacent to somite 2 was transplanted into anterior cranial mesoderm following extirpation of a region of host mesoderm (*Figure 5A*). Transplanted cells were incorporated into both dorsal and ventral mandibular arch or hyoid arch muscle (*Figure 5B,C''*). These muscles displayed normal innervation from the mandibular branch of the trigeminal nerve (*Figure 5C−C''*) and the facial nerve, respectively. Local cues appear sufficient to pattern myogenic lateral plate mesoderm from the cucullaris region and to promote mandibular or hyoid arch muscle development.

Next, more posterior lateral plate mesoderm, adjacent to somite 5, was transplanted heterotopically to mandibular arch mesoderm at stage 21 (*Figure 5D*). While transplanted cells were present among mandibular arch structures, in 9 of 10 larvae they did not incorporate into muscle (*Figure 5E*). Neither mandibular nor hyoid arch mesoderm appears sufficient to induce posterior non-myogenic lateral plate mesoderm to form muscle.

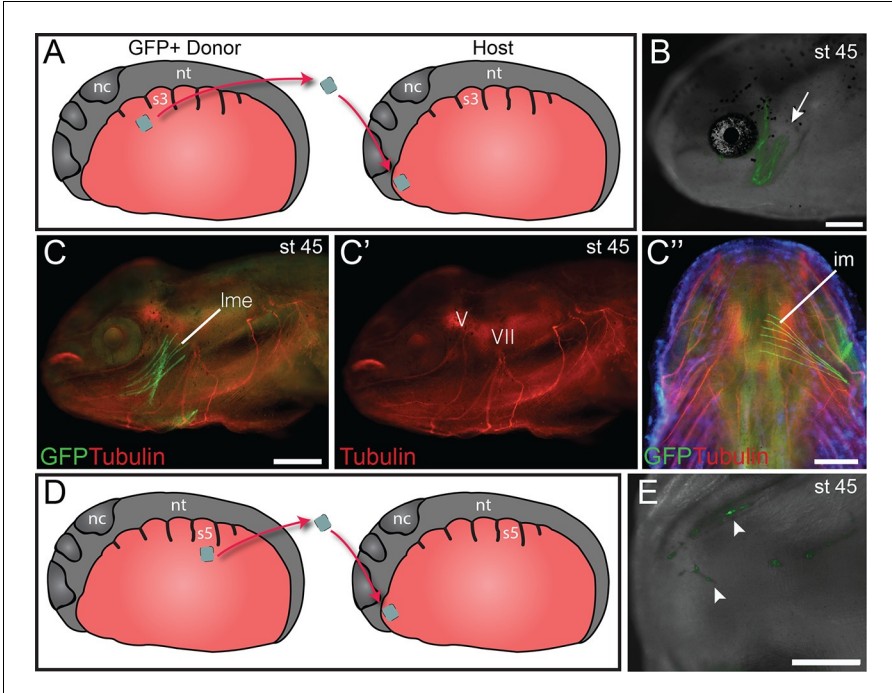

**Figure 5.** Heterotopic transplantation of lateral plate mesoderm. (**A**) Schematic depiction of a caudal-to-cranial heterotopic transplantation of lateral plate mesoderm from somite level 2 (donor) to mandibular arch mesoderm (host). Lateral views; anterior is to the left. nc, neural crest; nt, neural tube; s3, somite 3. (**B−C''**) Stage-45 larva following the heterotopic transplantation shown in (**A**). (**B**) GFP+ cells contribute to mandibular arch muscles (arrow). Lateral view; anterior is to the left. (**C**) Lateral plate mesoderm contributes to the levator mandibulae externus (lme). (**C'**) Innervation of the levator mandibulae externus by the mandibular branch of the trigeminal nerve (V) is normal. VII, facial nerve. (**C''**) The intermandibularis (im), a ventral mandibular muscle, is also labeled. Ventral view; anterior is to the top. (**D**) Schematic depiction of a caudal-to-cranial heterotopic transplantation of lateral plate mesoderm from somite level 5 to mandibular arch mesoderm. (**E**) Stage-45 larva following the heterotopic transplantation shown in (**D**). Ventral view; anterior is to the left. No muscle fibers are formed, but labeled cells contribute to cranial vasculature (arrowheads). Scale bars, 100 μm.

## Discussion

### Phylogenetic distribution of the cucullaris

We provide evidence from comparative morphology, embryonic fate mapping and gene expression that the cucullaris is a branchiomeric muscle in series with the gill levators and that it is stably conserved across gnathostomes as a link between head and trunk. Accordingly, we propose the fifth gill levator of the coelacanth to be homologous to the cucullaris, which, as in some sharks, rays and lungfish, attaches the pectoral girdle to the posteriormost gill bar (*Edgeworth, 1935*; *Greenwood and Lauder, 1981*). We regard this interpretation of data from coelacanth, viz., the cucullaris has reduced its dorsal attachment to the head/epaxial muscle fascia, more parsimonious than a previous interpretation that the cucullaris is absent and that a gill levator has entirely shifted its origin from the head to the pectoral girdle (*Greenwood and Lauder, 1981*; *Millot and Anthony, 1958*). In larval *Ichthyophis kohtaoensis*, a caecilian, the fourth gill levator is substantially larger than the anterior three levators (*Kleinteich and Haas, 2007*). The cucullaris could potentially develop from the caudalmost gill levator, as has been suggested in urodeles (*Edgeworth, 1935*; *Ziermann and Diogo, 2013*). The cucullaris of caecilians, lacking an insertion to the absent shoulder girdle, instead has evolved a patent connection between the otic capsule (as well as the dorsal trunk fascia) and fascia associated with ventral trunk musculature. It is uncertain if the anterior portion of the levator arcus branchiales complex, which inserts on the posteriormost gillbar, is also part of the cucullaris or instead represents only gill levators that did not degenerate following metamorphosis. In the former case, this unusual configuration might express an ancestral potential of the cucullaris to attach to the gill skeleton, and it evokes reports that the paired fin/limb apparatus has surprising developmental resemblance to the gill arches (*Gillis et al., 2009*).

The cucullaris has evolved to perform distinct functions in different lineages. In placoderms, for example, it may have depressed the head (*Trinajstic et al., 2013*). The morphology of the cucullaris in sharks and rays suggests the muscle in gnathostomes originates ancestrally from the pectoral girdle and inserts on two parts of the cranial skeleton: the posterior gill bar and the caudal region of the head. The connection to the gill arches was likely lost in early tetrapods (but possibly later reappeared in caecilians), while an alternate attachment to the clavicle evolved in some lineages. The cucullaris is purportedly absent in turtles and snakes, but recent work suggests that it may be present in both groups. In turtles, it has been proposed that the muscle originates on the shell (carapace), which incorporates parts of the pectoral girdle (*Lyson et al., 2013*). In snakes, the pectoral girdle is absent and the origin of the cucullaris has concomitantly shifted to the body wall (*Tsuihiji et al., 2006*).

### Cucullaris development in the axolotl

The cucullaris is located in a complex transition zone between head and trunk; in the axolotl, this complexity is reflected in the muscle's dual embryonic derivation from both somitic and cranial mesoderm. An origin from both the caudal branchial levator and somites was suggested in the spotted salamander, *Ambystoma maculatum*, based on serial sections and dissection (*Piatt, 1938*). Our finding that unsegmented mesoderm adjacent to the anterior somites forms the posterior gill-levator muscles, a laryngeal muscle, the levatores et depressores branchiarum and the cucullaris indicates that the posterior limit of cranial mesoderm is at somite 3. The presence of labeled cranial mesoderm cells in a laryngeal muscle in axolotl betrays the deep phylogenetic conservation of a relationship between the cucullaris and laryngeal muscles, which was revealed in a recent analysis demonstrating that mouse laryngeal muscles are clonally related to the trapezius and absent following mutation of the gene *Tbx1* (*Lescroart et al., 2015*). Moreover, expression of the genes *isl1* and *tbx1* in the gill-levator region suggests these muscles develop through the cranial muscle regulatory network, consistent with their classical anatomical classification as cranial muscles.

Our analysis of cranial mesoderm markers in axolotl provides additional evidence for genetic heterogeneity in cranial muscle development in anamniotes, which has been demonstrated in mouse and chicken (*Nathan et al., 2008*; *Dong et al., 2006*; *Kelly et al., 2004*; *Marcucio and Noden, 1999*). Surprisingly, our data reveal that differentiation of mandibular adductor muscles is present in amphibians at the level of gene expression. At stage 40, *isl1* is expressed in the superficial anterior adductor, while *lhx2* is expressed in the external adductor. In mouse, the LIM homeodomain gene

*Isl1* is required for normal second heart field (SHF) development and its expression in SHF progenitors is downregulated following differentiation (*Cai et al., 2003*). Genetic fate mapping in mouse demonstrates a large contribution of *Isl1*-positive cells to the ventral intermandibular muscle and the cucullaris (*Nathan et al., 2008*; *Theis et al., 2010*). In the axolotl at stage 40, *pitx2* is expressed in hyoid arch and tongue musculature but is not in the gill musculature. Taken together, these findings underscore the regionalization of developmental programs that underlies cranial muscle formation, both among pharyngeal arches and even within the mandibular adductor complex. Moreover, the broad phylogenetic diversity of the model species involved suggests that such regionalization may be an ancestral feature of tetrapod vertebrates that is retained in living taxa and may also exist in their piscine outgroups.

## Evolution of the head-trunk boundary

In the axolotl embryo, somites and pharyngeal arches are located at the same post-otic axial level, which is a basic feature of morphologically conservative vertebrates (*Kuratani, 1997*). Lateral plate mesoderm adjacent to somites 1 and 2 is located in the intermediate region between head and trunk and is important for morphogenetic movements associated with the migration of hypobranchial muscle progenitors (*Lours-Calet et al., 2014*). Moreover, lengthening of the amniote neck is associated with the caudal shift of the heart into the thorax (*Hirasawa et al., 2016*). The head-trunk interface at the paraxial level is marked by the path of circumpharyngeal neural crest cells as they migrate ventral to the occipital somites to form the circumpharyngeal ridge caudal to the pharynx (*Kuratani, 1997*). Specialized muscles occur at this paraxial level, including the trapezius/cucullaris and, in axolotls, the gill levators.

Our finding that the posterior gill-levator muscles and the cucullaris originate from cranial mesoderm adjacent to the first three somites supports categorization of the cucullaris as a branchiomeric muscle. Moreover, it may help explain why lateral plate mesoderm in the embryonic 'trunk' in chicken has myogenic capacity. Our fate-mapping data suggest that this mesoderm, which gives rise to the cucullaris in amniotes, is not a novel source of musculature, but instead is cranial mesoderm associated with the most posterior pharyngeal arch (5$^{th}$, 6$^{th}$ or 7$^{th}$, depending on species). We propose that, in the axolotl, somite 3 is the posterior limit of mesodermal contribution to cranial structures in both paraxial and lateral mesoderm (*Figure 6A*). In our heterotopic transplantations, cranial mesoderm that forms the cucullaris is able to follow the myogenic program of cranial muscles in the mandibular and hyoid arches. Although the chicken lacks many of the cartilages and muscles associated with the posterior pharyngeal arches in other tetrapods, it retains cucullaris progenitors in the same anatomical position as in the axolotl (*Figure 6B*).

The head-trunk boundary in the axolotl is congruent between cranial mesoderm and somitic mesoderm, but in the chicken (and probably other amniotes) the head-trunk boundary in somites is posterior to that in unsegmented cranial mesoderm (*Figure 6A–B*; *Couly et al., 1993*; *Piekarski and Olsson, 2014*; *Huang et al., 2000*). It remains to be determined whether this congruence, as seen in the axolotl, is the plesiomorphic condition for tetrapods. Heterotopic transplantations in chicken suggest that somitic mesoderm has greater regional plasticity than lateral plate mesoderm. Somites that contribute to the posterior skull are able to generate vertebrae when transplanted to a more posterior position, independent of *Hox* gene expression (*Kant and Goldstein, 1999*), whereas caudal cranial mesoderm that gives rise to the cucullaris is unable to generate muscle when transplanted to a more posterior location (*Theis et al., 2010*). It will be of interest to identify the mechanisms responsible for the incorporation of somites into the posterior skull during tetrapod evolution and to determine if the posterior limit of cranial mesoderm is less evolutionarily labile than somitic contribution to cranial structures.

## Materials and methods

### Contrast staining and micro-computed tomography (CT) scans

CT scans were prepared from anatomical specimens of *Hydrolagus* sp. (MCZ 164893), *Polypterus bichir* (MCZ 168418) and *Protopterus* sp. (MCZ 54055) from the Museum of Comparative Zoology at Harvard University, as well as *Typhlonectes natans* (YPM HERA 012618) and *Monodelphis domestica* (YPM MAM 10713) from the Yale Peabody Museum of Natural History at Yale University. *Latimeria*

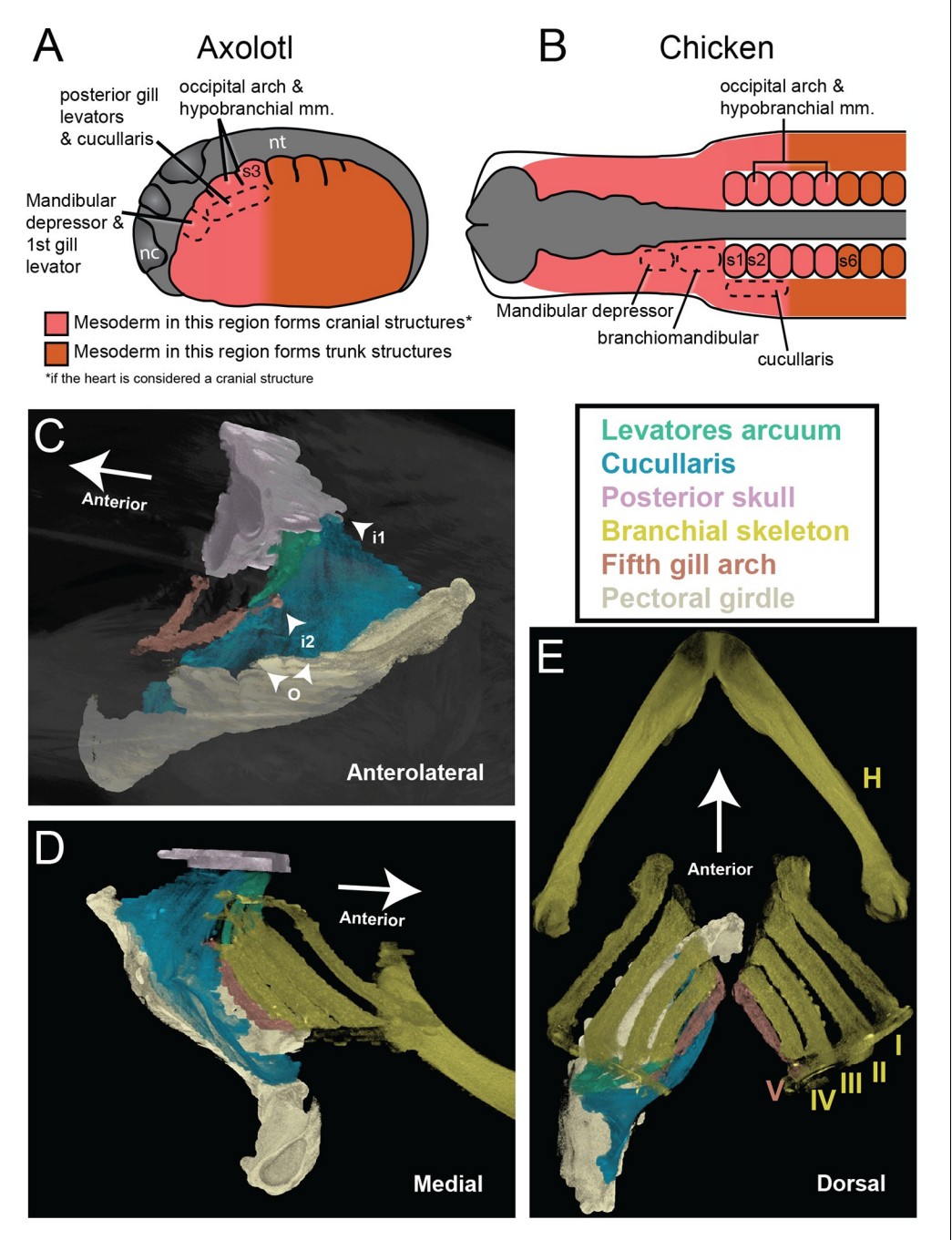

**Figure 6.** The cucullaris and the transition zone between the head and trunk. (**A**) In the axolotl embryo, the head-trunk boundary in unsegmented mesoderm is closely congruent with that in the somites. Paraxial and lateral mesoderm anterior to somite 3 form cranial structures (including the heart). The illustration depicts a stage-21 embryo with epidermis removed; anterior is to the left. Somite fate-mapping data are from *Piekarski and Olsson (2007)*; *Piekarski and Olsson (2014)*. (**B**) In the chicken, the axial level of the head-trunk boundary in somitic mesoderm is posterior to the border in unsegmented mesoderm. Somite fate-mapping data are from *Couly et al. (1993)* and *Huang et al. (1999)*; cucullaris data are from *Theis et al. (2010)*; mandibular depressors and branchiomandibular data are from *Noden (1983)* and *Evans and Noden (2006)*. (**C−E**) Contrast-stained CT images of the lungfish branchial skeleton, pectoral girdle, posterior skull, gill levators and cucullaris. All structures except the branchial skeleton are segmented on the left side only. The anterolateral view depicts only the fifth gill arch, with its attachment to the cucullaris; the body is rendered transparent. The lungfish cucullaris retains the ancestral tripartite attachment: origin from the pectoral girdle (o) and insertions on the posterior skull (i1) and fifth ceratobranchial (i2).

*chalumnae* (AMNH 32949h) was obtained from the American Museum of Natural History. For contrast staining, specimens were immersed in 5% Lugol solution in 70% ethanol for 7−10 d at room temperature. Specimens were washed in 70% ethanol for 2 d, changing solution daily. Three-dimensional images were taken using an XRA-002 microCT scanner (X-Tek; Tring, United Kingdom) at the Center for Nanoscale Systems at Harvard University. Reconstructions were performed with VGStudio Max 2.0 (Volume Graphics).

## Magnetic resonance imaging (MRI) data for coelacanth

MRI data for *Latimeria chalumnae* (SIO 75–347) from the Scripps Institution of Oceanography were obtained from the Digital Fish Library hosted by the University of California, San Diego, through the generosity of Lawrence Frank and Rachel Berquist.

## Fate-mapping in *Ambystoma mexicanum*

White mutant (dd), GFP+ white mutant and albino (aa) embryos of the Mexican axolotl (*Ambystoma mexicanum*) were obtained from the Ambystoma Genetic Stock Center at the University of Kentucky and from the Hanken laboratory breeding colony at Harvard University. Before grafting, embryos were decapsulated manually by using watchmaker forceps and then staged (*Bordzilovskaya et al., 1989*; *Nye et al., 2003*). Explants of unsegmented cranial mesoderm (stages 18–22) from donor embryos were grafted unilaterally into stage-matched hosts in place of comparable regions that had been extirpated. Stage-matched donors were of similar size and form. In heterotopic transplantations, anterior cranial mesoderm from regions that contribute to mandibular or hyoid arch musculature was partially extirpated in hosts. In one set of heterotopic experiments, GFP+ mesoderm adjacent to somite 2 was moved into either region 1 or region 2 of host anterior cranial mesoderm (integrating into either the mandibular or hyoid arch). In a second set of experiments, GFP+ mesoderm adjacent to somite 5 was transplanted into host anterior cranial mesoderm.

## Immunohistochemistry and sectioning

Fixation, embedding and sectioning were performed as previously described for *A. mexicanum* (*Sefton et al., 2015*). For GFP labeling, sections were incubated with rabbit polyclonal anti-GFP ab290 (1:2000; Abcam, Cambridge, MA), followed by AlexaFluor-488 goat anti-rabbit (1:500; Life Technologies, Carlsbad, CA). DAPI (0.1–1 μg/ml in PBS) was used to label cell nuclei. Some sections were stained with the skeletal muscle marker 12/101 monoclonal antibody (1:100; Developmental Studies Hybridoma Bank, Iowa City, IA). Additionally, desmin (1:100; Monosan, PS031; Uden, Netherlands) was used to label muscle in stage-40 embryos. Acetylated alpha-tubulin (1:100; Sigma, T6793; St. Louis, MO) was used to detect developing axons, followed by AlexaFluor-568 goat anti-mouse (1:500; Life Technologies, Carlsbad, CA).

## Optical projection tomography (OPT) of *Ambystoma mexicanum*

A specimen for OPT (*Sharpe et al., 2002*) was stained with 12/101 followed by AlexaFluor-568 goat anti-mouse as described above. Clearing and embedding were performed at the University of Washington, where the larva was dehydrated in ethanol, cleared in 1:2 benzyl alcohol/benzylbenzoate, and imaged with a Bioptonics 2100M scanner.

**Table 1.** Primer sequences.

| Transcript | Forward (5' to 3') | Reverse (5' to 3') | Product Size |
|---|---|---|---|
| lhx2 | AACAGTGACGCAAACAGTGG | TTGAAGCAGTTAGCGCAGAA | 755 bp |
| msc | ACCAGCAGACACCAAGCTCT | TGTGTCCTCCTCTGATGTGAA | 708 bp |
| pitx2 | AGATCGCCGTGTGGACTAAC | GGTGGTAGCGAGTTTTGGAA | 809 bp |
| tbx1 | GGAGTACGACCGAGATGGAA | ATGAAGCGCTGATGACAGTG | 688 bp |

## RNA in situ hybridization

In situ hybridization was performed on albino (aa) embryos. Antisense riboprobes were generated from the cloned fragment (DIG RNA labeling kit; Roche Diagnostics, Indianapolis, IN). In situ hybridization was carried out as previously described (*Henrique et al., 1995*), with an additional MAB-T wash overnight at 4°C (100 mM maleic acid, 150 mM NaCl, pH 7.5, 0.1% Tween 20). Hybridization was performed at 65°C. Primers are included in *Table 1*. Amplified PCR fragments were subcloned into the pCR II vector (Life Technologies).

## Acknowledgements

We are grateful to N Piekarski for her contribution of early transplantations and comments on the manuscript, and to C DeVane for expert animal care. We thank J Whited, B Haas and L Peshkin for the gift of sequences for *pitx2, tbx1* and *msc*. 12/101 antibody was from the Developmental Studies Hybridoma Bank developed under the auspices of the National Institute of Child Health and Human Development (NICHD) and maintained by the Department of Biology at the University of Iowa. We thank T Cox at the University of Washington Small Animal Tomographic Analysis Facility for assistance with OPT. This work was supported by a Graduate Women in Science Vessa Notchev Fellowship to ES.

## Additional information

### Funding

| Funder | Author |
| --- | --- |
| Sigma Delta Epsilon-Graduate Women in Science | Elizabeth M Sefton |

The funders had no role in study design, data collection and interpretation, or the decision to submit the work for publication.

### Author contributions

EMS, B-ASB, Conception and design, Acquisition of data, Analysis and interpretation of data, Drafting or revising the article; ZM, Acquisition of data, Analysis and interpretation of data, Drafting or revising the article, Designed research; JH, Conception and design, Analysis and interpretation of data, Drafting or revising the article

### Author ORCIDs

Elizabeth M Sefton, http://orcid.org/0000-0001-6481-612X
Bhart-Anjan S Bhullar, http://orcid.org/0000-0002-0838-8068
Zahra Mohaddes, http://orcid.org/0000-0002-3766-5896
James Hanken, http://orcid.org/0000-0003-2782-9671

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
