## [Decision Letter]

Thank you for submitting your work entitled "Evolution of the head-trunk interface in tetrapod vertebrates" for peer review at *eLife*. Your submission has been favorably evaluated by Diethard Tautz as Senior editor, a Reviewing editor, and two reviewers.

This paper explores the head-trunk transition in vertebrates. It provides a review of classical ideas as well as recent developmental and molecular studies that impact this subject, most specifically the contrast in genetic regulation of cranial versus trunk muscles. The classical literature holds some controversy about the branchiomeric origin of the cucullaris muscle and its phylogenetic distribution. That muscle and its homologues are the major focus of this study. CT scans of numerous taxa are included to argue that the cucullaris is a universal feature of gnathostomes. The authors present orthotopic transplant experiments in axolotl to build a fate map confirming that the posterior gill levators and the cucullaris in this amphibian are derived from lateral plate mesoderm adjacent to somites 1-3. They also preform heterotopic transplants of LPM from different axial levels to test for myogenic properties in response to cranial versus trunk signaling environments. They demonstrate the ability of anterior LPM cells to respond to cranial signals and differentiate as muscle where as in contrast, LPM posterior to somite 3 does not respond in the same site, and fails to differentiate as muscle. The authors conclude that the true head/trunk boundary occurs in both somites and LPM behind somite 3 in axolotl.

The timing of this manuscript is good. Several other studies have requested this kind of data from an anamniote species to properly discuss the evolution of neck development. This is an important confirmation for an important idea. However, a number of points need to be addressed by the authors.

1) The manuscript is uneven, and the CT data seems crammed in. That data should be worked over to make a more fluid seam between the anatomy and embryology. The sentence “Based on these data, it is unknown whether the lateral plate origin of the cucullaris is the result of a posterior shift of the head myogenic program or instead represents head mesoderm that spread caudally into the region adjacent to the anterior somites”, asks a specific question, but this gets lost in subsequent discussion and CT scans- development and anatomy are not well integrated, giving the impression that there are two papers cobbled together. This requires revision.

2) What are the criteria for identifying these muscles in the CT scans?

These data are surprisingly underwhelming. If the technique is used in order to avoid destruction of rare specimens during dissection, maybe they would seem worthwhile. What is described in the text for the coelacanth is not clearly visible in Figure 1.

3) Neural crest origin of connective tissue in the spinotrapezius has not been confirmed. Other studies show a lateral plate lineage for this CT (Durland et al. 2008)

4) Schematics should be added to show surgeries in Figure 2 & 3.

5) The fate of somites 1, 2 & 3 in axolotl needs to be mentioned early so the axial level coincidence of somites and LPM makes sense. For instance, the last two lines of the Abstract could appear to contradict each other without the fact that somites 1-2 are occipital.

6) The last sentence of the subsection “Phylogenetic distribution of the cucullaris “presumably refers to homology of nuchal bone of carapace with the cleithrum, this is controversial and should be qualified.

[Editors' note: further revisions were requested prior to acceptance, as described below.]

Thank you for resubmitting your work entitled "Evolution of the head-trunk interface in tetrapod vertebrates" for further consideration at *eLife*. Your revised article has been favorably evaluated by Diethard Tautz as Senior editor and a Reviewing editor. The manuscript has been greatly improved but there is one remaining issue, that was not emphasised in the previous reviews, that needs to be addressed before acceptance, as outlined below:

The section on "Molecular regionalization of mandibular and hyoid arch muscles" presents very nice data but there is no reference to this section in the Introduction or the Discussion. We would ask you to better integrate it into the rest of the manuscript.

---

## [Author Response]

*The timing of this manuscript is good. Several other studies have requested this kind of data from an anamniote species to properly discuss the evolution of neck development. This is an important confirmation for an important idea. However, a number of points need to be addressed by the authors. 1) The manuscript is uneven, and the CT data seems crammed in. That data should be worked over to make a more fluid seam between the anatomy and embryology. The sentence “Based on these data, it is unknown whether the lateral plate origin of the cucullaris is the result of a posterior shift of the head myogenic program or instead represents head mesoderm that spread caudally into the region adjacent to the anterior somites”, asks a specific question, but this gets lost in subsequent discussion and CT scans- development and anatomy are not well integrated, giving the impression that there are two papers cobbled together. This requires revision.*

We’ve modified the Introduction to clarify the relationship between our anatomical and developmental descriptions, viz., each represents a source of data that can be used to evaluate the branchiomeric identity of the cucullaris. We have enlarged individual panels in Figure 1 to make them appear less crowded and to allow features on individual panels to be discerned more easily. In addition, CT data from the axolotl has been incorporated into Figure 2 to create a more effective transition between CT and fate-mapping data. We added an optical projection tomography scan of a juvenile (stage 53) axolotl to the same figure to more readily contrast the morphology of the cucullaris between juveniles and adults. The question in the Introduction has been moved to a new paragraph to emphasize its importance, and it is now addressed more explicitly in the Results (beginning of the section, “Mesoderm adjacent to anterior somites forms the cucullaris and gill levators”).

*2) What are the criteria for identifying these muscles in the CT scans? These data are surprisingly underwhelming. If the technique is used in order to avoid destruction of rare specimens during dissection, maybe they would seem worthwhile. What is described in the text for the coelacanth is not clearly visible in Figure 1.* We added several sentences to the Introduction that explain the utility of CT/MRI scans for examining the cucullaris, e.g., they accurately depict the muscle’s three-dimensional morphology without damaging rare specimens (especially the adult and pup coelacanth). The criteria we use for muscle identification include the specific attachment points (origin and insertion) as well as overall morphology (e.g., shape) and fiber orientation. In the case of the coelacanth, we consider it more parsimonious to conclude that the cucullaris has reduced its dorsal attachment to the head (while maintaining the ancestral connection between the posteriormost branchial bar and pectoral girdle) than it is to propose that a gill levator has gained an entirely new origin from the pectoral girdle. We have made these points more explicit in the Results and Discussion.

*3) Neural crest origin of connective tissue in the spinotrapezius has not been confirmed. Other studies show a lateral plate lineage for this CT (Durland* et al.

*2008)*

We have added a sentence that notes the existence of conflicting claims regarding the source of connective-tissue components of the trapezius in mouse.

*4) Schematics should be added to show surgeries in Figure 2 & 3.*

Schematics have been added to Figure 2 and 3. We also divided Figure 2 and Figure 3 into three separate panels so that each could fit on a single page (now Figure 2–Figure 4). As such, there are now 6 primary figures total instead of 5.

*5) The fate of somites 1, 2 & 3 in axolotl needs to be mentioned early so the axial level coincidence of somites and LPM makes sense. For instance, the last two lines of the Abstract could appear to contradict each other without the fact that somites 1-2 are occipital.*

We added three sentences to the second paragraph of the Introduction to more effectively introduce this topic. We also changed the second-to-last sentence of the Abstract to “Accordingly, LPM adjacent to the occipital somites should be regarded as posterior cranial mesoderm.”

6) The last sentence of the subsection “Phylogenetic distribution of the cucullaris“, presumably refers to homology of nuchal bone of carapace with the cleithrum, this is controversial and should be qualified.

We qualified this statement as follows: “In turtles, it has been proposed that the muscle originates on the shell (carapace), which incorporates parts of the pectoral girdle (Lyson et al. 2013).”

[Editors' note: further revisions were requested prior to acceptance, as described below.]

The manuscript has been greatly improved but there is one remaining issue, that was not emphasised in the previous reviews, that needs to be addressed before acceptance, as outlined below: There is one part that now stands out which we did not draw your attention to earlier. The section on "Molecular regionalization of mandibular and hyoid arch muscles" presents very nice data but there is no reference to this section in the Introduction or the Discussion. We would ask you to better integrate it into the rest of the manuscript.

We have addressed the above request by revising the Introduction and the Discussion in the following ways:

In the first paragraph of the Introduction, we added a sentence that indicates the importance of differential genetic control in cranial muscles based on previous research in mouse.

Later in the Introduction (last paragraph), we broadly summarize our results from this subsection.

In the Discussion (subsection “Cucullaris development in the axolotl“, last paragraph), we modified an existing paragraph to highlight and more clearly synthesize these results.

We also made a few minor edits to clarify points and correct awkward syntax (Introduction, Discussion). We updated a reference that was previously ‘In press’ (Hirasawa, Fujimoto and Kuratani, 2016).